# Application of a Loop-Mediated Isothermal Amplification (LAMP) Assay for the Detection of *Listeria monocytogenes* in Cooked Ham

**DOI:** 10.3390/foods12010193

**Published:** 2023-01-01

**Authors:** Alfonsina Fiore, Ida Treglia, Gianni Ciccaglioni, Marco Francesco Ortoffi, Antonietta Gattuso

**Affiliations:** Dipartimento di Sicurezza Alimentare, Nutrizione e Sanità Pubblica Veterinaria—Istituto Superiore di Sanità, 00161 Roma, Italy

**Keywords:** *Listeria monocytogenes*, rapid detection, colorimetric loop mediated isothermal amplification (colorimetric-LAMP), real-time LAMP PCR

## Abstract

Changing eating habits and rising demand of food have increased the incidence of foodborne diseases, particularly in industrialized countries. In this context, contaminated ready-to-eat food (RTE) may be a vehicle for the transmission of *Listeria monocytogenes* (*L. monocytogenes*), a foodborne pathogen responsible of listeriosis, a severe infectious disease involving humans and animals. It would be useful to have rapid detection methods to screen the presence of *L. monocytogenes* in food. In this study, a colorimetric Loop-mediated isothermal amplification (LAMP) assay was applied to the detection of *L. monocytogenes* in 37 experimentally contaminated RTE meat samples. The LAMP primers consisted of a set of six primers targeting eight regions on the *hly*A gene; the assay was carried out in 30 min at 65 °C in a water bath. Amplification products were visualized by color change assessment. The results of colorimetric LAMP assays based on the *hly* gene obtained in this study were compared to microbiological cultural methods, real-time PCR and real-time LAMP PCR, which show 100% specificity and sensitivity. These data suggest that colorimetric LAMP assays can be used as a screen to detect *L. monocytogenes* in ready-to-eat meat food.

## 1. Introduction

*Listeria monocytogenes* (*L. monocytogenes*) is one of major foodborne pathogens responsible for listeriosis, a severe infectious disease both in humans and animals. Although the occurrence of listeriosis is low, the hospitalization and fatality rates are high [1]. Unlike many other common foodborne pathogens, *L. monocytogenes* can survive and multiply at acidic values of pH, high concentrations of NaCl, and at low temperatures [2], increasing the risk of contamination in certain types of food such those with a long shelf-life under refrigeration and foods that are consumed without being subjected to heat treatment. Among these, ready-to-eat foods (RTE), including raw vegetables, dairy and meat products, and seafood [3,4,5], can become contaminated during processing. The pathogen is able to multiply to dangerous levels during distribution and storage, becoming a risk since the foods are consumed without the need for cooking or other processing able to eliminate or reduce the microorganism [6]. Traditional culture-based methods for isolating and enumerating *L. monocytogenes* from food samples generally involve microbiological techniques [7]. Although widely used, these methods require laborious steps and are very time-consuming (5–7 days for a positive result). In recent years, some rapid molecular methods, such as real-time PCR assays, have been developed for the detection of *L. monocytogenes* and to increase speed and sensitivity [8]. Nonetheless, these assays require dedicated equipment, which is rather expensive and complicated, as well as qualified personnel. The loop-mediated isothermal amplification (LAMP) assay is based on the isothermal amplification of DNA by a thermostable polymerase with “strand displacement” activity, obtained from *Bacillus stearothermophilus* (Bst polymerase) and on a set of six different primers, specially designed, which are specific for the species to be identified; no special apparatus is needed [9]. A simple laboratory water bath or a heating block is adequate for generating the single temperature needed for the synthesis of DNA targets that can be generally observed by a visual assessment of turbidity, color change, or by the addition of fluorescent reagents [3,10,11]. In this work, we applied a simple, rapid, and cost-effective method, based on the “colorimetric LAMP assay”, for detection of *L. monocytogenes* in food products and which can be performed “*in the field*” and used for screening. Specifically, colorimetric LAMP was used to detect *L. monocytogenes* in experimental contaminated ready-to-eat meat food, specifically, cooked ham. The results were compared with a microbiological culture based methods [7] and a real time assay specific for *L. monocytogenes* [12]. In addition, a rapid real-time LAMP PCR was applied to confirm colorimetric LAMP results.

## 2. Materials and Methods

### 2.1. Bacterial Strains and Culture Conditions

Eighteen bacterial isolates (Table 1) obtained both from ATCC and other collections were used for LAMP and real-time LAMP PCR assay specificity tests. *L. monocytogenes* ATCC 13932 and *Listeria innocua* (*L. innocua*) ATCC 33090 were also used as the positive and negative controls, respectively. The strains were grown overnight in 10 mL of Tryptone Soya Broth (TSB—Biolife Italiana s.r.l.) at 37 °C before use.

### 2.2. Bacterial DNA Extraction

DNA was extracted from an overnight culture of microorganisms grown in TSB. Briefly, 2 ml of these culture were pelleted by centrifugation at 14,000 rpm for 10 min at 4 °C, and pellet was re-suspended in 1 mL of sterile deionized nanopure water. After centrifugation at 14 000 rpm at 4 °C for 10 min, the pellet was suspended in 200 µL of 6% Chelex^®^ 100 (Sigma Aldrich) and heated at 100 °C for 8 min in a thermoblock. Immediately, the test tube was placed on ice for 1 min and after centrifuged at 14,000 rpm for 10 min at 4 °C. About 100 µL of the DNA supernatant was carefully transferred to a new sterile tube to be used as a template for the planned amplification reactions.

### 2.3. Primers

A set of six LAMP primers, including two inner primers (forward inner primer [FIP] and backward inner primer [BIP]), two outer primers (F3 and B3), and two loop primer (LF and LB) that recognized eight distinct regions in the target DNA (gene *hly*A), was used [13] both for colorimetric LAMP and real-time LAMP PCR assays. For real-time PCR we used primer and probe targeting for gene *hly*A [14]. All primers and probes were synthesized by Eurofins Genomics Italy S.r.l. and are listed in Table 2.

### 2.4. Colorimetric LAMP Assay

The colorimetric LAMP was performed in a 25 µL final volume solution for each sample. The mix consisted of 12.5 µL WarmStart Colorimetric LAMP 2X Master Mix (New England Biolabs, Ipswich, MA, USA), 2.5 µLof LAMP primer mix (final concentration:1.6 μM LAMP inner primers [FIP and BIP], 0.2 μM LAMP outer primers [F3 and B3], 0.4 μM LAMP loop primers [LF and LB]), 2 µL of DNA target, and 8 µL of sterile nuclease-free water. *L. monocytogenes* ATCC 13932 and *L. innocua* ATCC 33090 were used as positive and negative controls, respectively, and were included in each LAMP assay. In addition, a negative control reaction was performed using sterile nuclease-free water instead of the DNA target. The colorimetric LAMP mix, placed in a 0.5 mL reaction tube, was heated at 65 °C for 30 min in a water bath and visually examined at the end time in order to visualize the color change from pink to yellow. All positive amplifications appeared yellow, while negative controls were pink. If the color change was faint, incubation was prolonged for an additional 10 min. The results can be recorded, of course, or simply photographed or kept leaving the reaction tubes at room temperature.

### 2.5. Colorimetric LAMP Assay: Specificity and Sensitivity Determination

In order to determine the specificity of LAMP assay, the LAMP reaction was performed under the conditions described above using DNA templates from the 18 bacterial type strains (Table 1).

The sensitivity (limit of detection, LOD) of the LAMP assays was assessed using DNA extracted from *L. monocytogenes*. Briefly, an overnight culture of *L. monocytogenes* ATCC 13932 in TSB was serially diluted with MRD to get a final concentrations ranging from 5.5 to 5.5 × 10^7^ CFU/mL. Then, 2 mL of each dilution was used for DNA extraction and subjected to colorimetric LAMP assay as described above. The LOD was expressed as the lowest concentration of *L. monocytogenes* (CFU/mL) that gave a positive result in all of the three replicates.

### 2.6. Real-Time LAMP PCR

To confirm the specificity of the colorimetric LAMP assay products, a real-time LAMP PCR assay was applied [15]. The mix reaction consisted of 12.5 µL of WarmStart LAMP Kit (New England Biolabs, Ipswich, MA, USA), 0,5 µL of fluorescent dye (New England Biolabs, Ipswich, MA, USA), 2.5 µL LAMP primer mix (final concentration:1.6 μM LAMP inner primers [FIP and BIP], 0.2 μM LAMP outer primers [F3 and B3], 0.4 μM LAMP loop primers [LF and LB]), 2 µL of DNA target, and 7.5 µL of sterile nuclease-free water in a final volume of 25 µL. A positive control *L. monocytogenes* ATCC 13932 and a negative control *L. innocua* ATCC 33090 were included in each real-time LAMP PCR assay. In addition, a negative control reaction was performed using sterile nuclease-free water instead of the DNA target. The real-time LAMP PCR mix was placed in a 96-well plate and subjected to amplification by the following thermal profile: 30 cycles at 65 °C for 1 min, one cycle at 95 °C for 1 min, 55 °C for 30 s and 95 °C for 30 s in a Stratagene 3005Xp Thermal Cycler (Agilent).

### 2.7. Real-Time PCR

The real-time PCR was performed in a total volume of 25 µL containing 12.5 µL of 1X Master MIX (Quiagen), 0.075 µL of *hly*F primer (final concentration 300 nM), 0.075 µL of *hly*R primer (final concentration 300 nM), 0.025 µL of *hly*P probe (final concentration 100 nM), 8.3 µL of sterile nuclease-free water, and 3 µL of DNA target. A positive control *L. monocytogenes* ATCC 13932 and a negative control *L. innocua* ATCC 33090 were included in each real-time PCR assay; in addition, a negative control reaction was performed using sterile nuclease-free water instead of the DNA target. The assay was carried out through an initial denaturation at 95 °C for 15 min, followed by 40 cycles of denaturation at 95 °C for 15 s, and primer annealing at 63 °C for 60 s [8] in a Stratagene 3005Xp Thermal Cycler (Agilent Technologies Italia S.p.A).

### 2.8. Food Samples, Experimental Contamination, and Procedure

Briefly, 925 grams of cooked ham, purchased in a local market, was divided into 3 portions: (i) a portion of 300 g was experimentally contaminated with an overnight culture of *L. monocytogenes* ATCC 13932 to obtain a final concentration of 10^4^ CFU/g; (ii) a portion of 325 g was experimentally contaminated with an overnight culture of *L. monocytogenes* ATCC 13932 to obtain a final concentration of 10^6^ CFU/g; (iii) a portion of 300 g was experimentally contaminated with an overnight culture of *Escherichia coli* ATCC 25922 and *Staphylococcus aureus* ATCC 25923 to obtain a final concentration of 10^4^ CFU/g (negative controls). Four different methods were compared to detect *L. monocytogenes* in all the samples: cultural microbiological method [7], real-time PCR [8], colorimetric LAMP assay, and real-time LAMP PCR.

From each contaminated portion, 25-gram samples were placed in sterile bags (Nasco Whirl-Pak, VWR International), yielding 37 samples; in each bag was added 225 mL of sterile non-selective primary enrichment broth (Half Fraser broth, Biolife Italiana S.r.l.).

All the 37 bagged samples were crushed and homogenized in a stomacher blender 400 circular (P.B.I.) and incubated at 30 °C for 24 h, as required by the cultural microbiological method [7]. After the pre-enrichment step, 2 mL of each sample was subjected to DNA extraction as previously described, and detection of *L. monocytogenes* was carried out as required by colorimetric LAMP, real-time PCR, and real-time LAMP PCR assays. In parallel, all samples were analyzed by cultural microbiological ISO 11290-1 method.

## 3. Results

### 3.1. Specificity and Sensitivity of Colorimetric LAMP

The specificity test performed by submitting all *L. monocytogenes* strains to colorimetric LAMP revealed that a positive result (color change from pink to yellow) was observed within 30 min, as planned (Figure 1); no color change was obtained both for all non-*L. monocytogenes* strains (Figure 2) and for other strains (Figure 3).

The analytical sensitivity (LOD) of the colorimetric LAMP assay, assessed using 10-fold serial dilutions of *L. monocytogenes* ATCC 13932 pure culture, equals 55 CFU/mL (Figure 4).

### 3.2. Specificity of Real-Time LAMP PCR

The specificity test of real-time LAMP PCR was carried out using strains listed in Table 1. Results are shown in Appendix A and Table 3, Table 4 and Table 5; only *L. monocytogenes* strains were amplified (Appendix A; Table 3), no amplifications were obtained for either non-*L. monocytogenes* (Appendix A; Table 4) or other strains (Appendix A; Table 5).

### 3.3. Detection of L. monocytogenes in Artificially Contaminated Samples

According to the experimental conditions, cultural the microbiological method found that 25 test samples were positive, while 12 test samples were negative. Colorimetric LAMP, real-time LAMP PCR, and real-time PCR obtained the same results after only 24 h. The results of colorimetric LAMP assay for detection of *L. monocytogenes*, based on the *hly* gene detection, compared to microbiological cultural method, real-time LAMP PCR and real-time PCR showed 100% specificity and sensitivity.

## 4. Discussion

The LAMP assay provides a tool for establishing a rapid diagnostic technique based on the molecular amplification of DNA. In this study, a colorimetric loop-mediated isothermal amplification (LAMP) assay was applied for the detection of *L. monocytogenes* from RTE meat food, specifically from cooked ham, which is the one most likely to be contaminated with this pathogen. Many studies performed LAMP assays for detection of *L. monocytogenes* [3,10,11,13,16,17]. Generally, amplified products have been evaluated through calcein, manganous ions, SYBR Green, as well as agarose gel electrophoresis. The last of these assays has several drawbacks, such as the use of highly toxic ethidium bromide, which is hazardous to health and the environment. On the other hand, calcein and SYBR Green, which are employed to avoid the use of dangerous ethidium bromide, require equipment for observing illumination, which may be an inconvenience for the user. Moreover, changes in turbidity mainly due to the white, focal phosphatase precipitate generated during the reaction can be difficult to detect at low levels. The colorimetric LAMP assay applied in this study, unlike the other LAMP-based methods, is more reliable because the use of a ready-to-use reaction mixture minimizes errors due to the use of reagents added separately to the reaction. In addition, visualization of the amplification product within the same reaction tube reduces the risk of contamination of the reaction product. Therefore, with the same efficacy compared to other LAMP assay (13), the colorimetric LAMP assay used in this study is more user friendly because no sample handling is required after DNA amplification. It is a sensitive and simple one-step method that can also be used in the field, only needing a thermostatically controlled bath. This method could be used for health management applications involving the fast identification of *L. monocytogenes* outbreaks to prevent foodborne diseases, decreasing hospitalization and mortality rates from human illness. Moreover, this method would be also recommended as screening to control *L. monocytogenes* spread along food production chain and in the food market and catering environment, reducing economic loss. At the same time, it would simplify the surveillance of the foodborne pathogens instead of time-consuming microbiological cultural methods and expensive molecular biology techniques. It can also be applied for the detection of *L. monocytogenes* from other RTE foods. In this regard, it would be appropriate in the future perform further studies to apply the colorimetric LAMP assay to detection of *L. monocytogenes* in naturally contaminated samples as they have a low concentration of the pathogen. Finally, our results show that colorimetric LAMP assays can be considered as an effective point-of-care test for the detection of *L. monocytogenes* in RTE meat-based food, easily usable also in developing countries. Recently, this aspect has been highlighted as an innovative tool for rapid diagnosis of pathogens [18].

## Figures and Tables

**Figure 1 foods-12-00193-f001:**
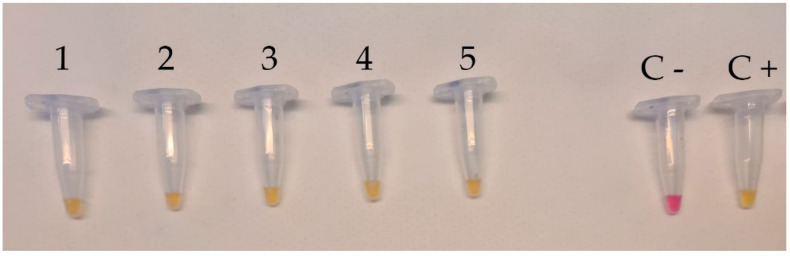
Specificity test of *L. monocytogenes* by colorimetric LAMP assay using DNA isolated from *L. monocytogenes* ATCC 13932 (Lane 1), *L. monocytogenes* ATCC 35152 (Lane 2), *L. monocytogenes* ATCC 7644 (Lane 3), *L. monocytogenes* ATCC 19111 (Lane 4), *L. monocytogenes* ATCC 19115 (Lane 5), negative control (Lane C−), and positive control (Lane C+). Each assay was carried out in triplicate.

**Figure 2 foods-12-00193-f002:**
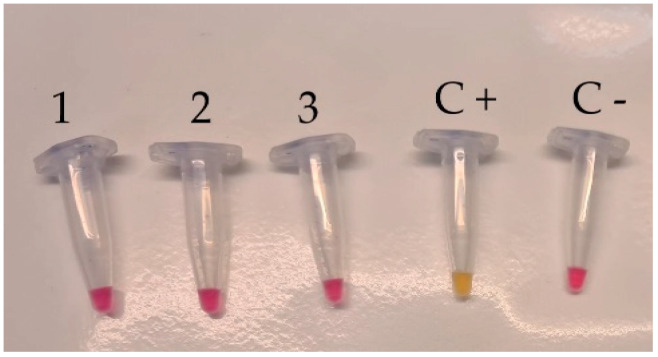
Specificity test of *L. monocytogenes* by colorimetric LAMP assay using DNA isolated from *L. innocua* (Lane 1), *L. ivanovii* (Lane 2), *L. seeligeri* (Lane 3), positive control (Lane C+), and negative control (Lane C−). Each assay was carried out in triplicate.

**Figure 3 foods-12-00193-f003:**
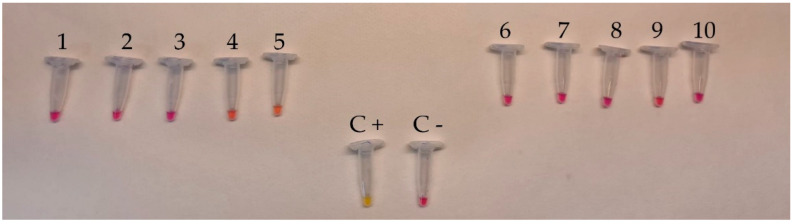
Specificity test of *L. monocytogenes* by colorimetric LAMP assay using DNA isolated from *Escherichia coli* ATCC 25922 (Lane 1), *Salmonella Enteritidis* ATCC 13076 (Lane 2), *Salmonella Typhimurium* ATCC 14028 (Lane 3), *Staphylococcus aureus* ATCC 25923 (Lane 4), *Enterococcus faecalis* ATCC 29212 (Lane 5), *Enterobacter aerogenes* ATCC 13048 (Lane 6), *Pseudomonas aeruginosa* ATCC 9027 (Lane 7), *Bacillus cereus* ATCC 6633 (Lane 8), *Citrobacter freundii* ATCC 8090 (Lane 9), *Proteus vulgaris* ATCC 13315 (Lane 10), positive control (Lane C+), and negative control (Lane C−). Each assay was carried out in triplicate.

**Figure 4 foods-12-00193-f004:**
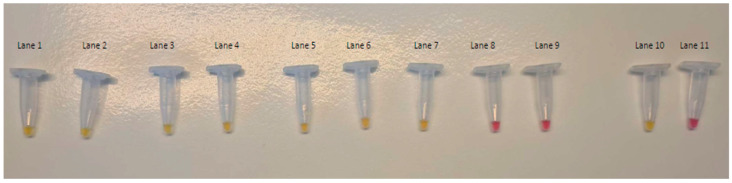
Sensitivity test of colorimetric LAMP assays. Lanes 1 to Line 9: DNA template extracted from *L. monocytogenes* ATCC 13932 from 5.5 × 10^7^ CFU/mL to 5.5 × 10^−1^ CFU/mL. Lane 10: positive control. Lane 11: negative control.

**Table 1 foods-12-00193-t001:** Bacterial isolates for specificity test.

Bacterial Strains	Source
*L. monocytogenes*	……………
*L. monocytogenes*	* ATCC 13932
*L. monocytogenes*	ATCC 35152
*L. monocytogenes*	ATCC 7644
*L. monocytogenes*	ATCC 19111
*L. monocytogenes*	ATCC 19115
Listeria spp	……………
*Listeria innocua*	ATCC 33090
*Listeria ivanovii*	ATCC 19119
*Listeria seeligeri*	** CLISS 11
Others strains	……………….
*Escherichia coli*	ATCC 25922
*Salmonella Enteritidis*	ATCC 13076
*Salmonella Typhimurium*	ATCC 14028
*Staphylococcus aureus*	ATCC 25923
*Enterococcus faecalis*	ATCC 29212
*Enterobacter aerogenes*	ATCC 13048
*Pseudomonasaeruginosa*	ATCC 9027
*Bacillus cereus*	ATCC 6633
*Citrobacter freundii*	ATCC 8090
*Proteus vulgaris*	ATCC 13315

* ATCC: strains purchased from American Type Culture Collection; ** CLISS: strain from Istituto Superiore di Sanità Culture Collection.

**Table 2 foods-12-00193-t002:** LAMP, real-time LAMP PCR, and real-time PCR primers used in this study.

Primer	Primer Sequence 5′-3′	References
LAMP/real-time LAMP PCR assay		
FIP_14 A	TCGCTCCAGTTTTTATGTTGAACACCTTGGGATGAARTAAATTATGATCC	[13]
BIP_12	AGCAAGCTAGCTCATTTCACATAGCGTAAACATTAATATTTCTCGC	[13]
F3_12	GGAGGMTACGTTGCTCAA	[13]
B3_12	AAGCTAAACCAGTGCATTC	[13]
LF_12	ACTTCCATTKCTTTA	[13]
LB_12	CGTCCATCTATTTGCCAGGTAAC	[13]
Real-time PCR		
*hly*F	CATGGCACCACCAGCATCT	[14]
*hly*R	ATCCGCGTGTTTCTTTTCGA	[14]
*hly*P	6-FAM- CGCCTGCAAGTCCTAAGACGCCA-BHQ1	[14]

**Table 3 foods-12-00193-t003:** Specificity of real-time LAMP PCR assay using DNA isolated from *L. monocytogenes* strains. Strain details of results.

Strains	Cycle Threshold	Melting Temperature
*L. monocytogenes* ATCC 13932	19.09	83 °C
*L. monocytogenes* ATCC 35152	18.90	83 °C
*L. monocytogenes* ATCC 7644	18.29	83 °C
*L. monocytogenes* ATCC 19111	17.96	83 °C
*L. monocytogenes* ATCC 19115	17.88	83 °C
Positive control	18.25	83 °C
Negative control	No Ct	79.7 °C

**Table 4 foods-12-00193-t004:** Specificity of real-time LAMP PCR assay using DNA isolated from non-*L. monocytogenes* strains. Strain details of results.

Strains	Cycle Threshold (Ct)	Melting Temperature
Listeria innocua ATCC 33090	No Ct	79.7 °C
Listeria ivanovii ATCC 19119	No Ct	79.7 °C
Listeria seeligeri CLISS 11	No Ct	79.7 °C
Positive control	19.13	83 °C
Negative control	No Ct	79.7 °C

**Table 5 foods-12-00193-t005:** Specificity of real-time LAMP PCR assay using DNA isolated from other strains. Strain details of results.

Strains	Cycle Threshold (Ct)	Melting Temperature
*Escherichia coli* ATCC 25922	No Ct	79.2 °C
*Salmonella Enteritidis* ATCC 13076	No Ct	79.2 °C
*Salmonella Typhimurium* ATCC 14928	No Ct	79.2 °C
*Staphylococcus aureus* ATCC 25923	No Ct	79.7 °C
*Enterococcus faecalis* ATCC 29212	No Ct	79.7 °C
*Enterobacter aerogenes* ATCC 13048	No Ct	79.2 °C
*Pseudomonas aeruginosa* ATCC 9027	No Ct	79.7 °C
*Bacillus cereus* ATCC 6633	No Ct	79.7 °C
*Citrobacter freundii* ATCC 8090	No Ct	79.7 °C
*Proteus vulgaris* ATCC 13315	No Ct	79.7 °C
Positive control.	18.94	83 °C
Negative control	No Ct	79.7 °C

## Data Availability

No new data were created or analyzed in this study. Data sharing is not applicable to this article.

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
