# Peer review of "Application of a Loop-Mediated Isothermal Amplification (LAMP) Assay for the Detection of Listeria monocytogenes in Cooked Ham"

_foods, 2023, doi:10.3390/foods12010193_

Round 1

Reviewer 1 Report (New Reviewer)

The manuscript presents the use of previously published LAMP assay technique for the detection of Listeria monocytogenes in the cooked ham.

As a novelty, the authors have introduced the colorimetric visualization results for detecting LAMP products. Furthermore, they compared the efficacy of LAMP with microbiological cultural method, Real-Time PCR and Real- Time LAMP PCR in cooked ham spiked with L. monocytogenes (low and high concentration), E. coli and S.aureus.

In my opinion using communication as a form of article is a very good solution due to relatively low novelty of the study and low number of tested samples. The article is generally well written but honestly the scientific value is not very high.

I have a few issues for the authors which should be improved/considered:

1.       Could the authors merge the parts Food samples and artificial contamination and Procedure into one and describe step by step how was the sample prepared and spiked with microorganisms. It is not clear in the form performed by the authors.

2.       The number of tested samples is very low for comparison of the methods, especially that no differences between efficacy was obtained. Could the authors provide the results from validation of the methods?

3.       According to the authors novelty of the study was based on using LAMP for identification of L. monocytogenes DNA of in raw and ready-to-eat meat. Could the authors compared the efficacy of their “new” colorimetric LAMP with results of Wu et al (2014)? It will be needed in the discussuion part.

Author Response

1. Could the authors merge the parts Food samples and artificial contamination and Procedure into one and describe step by step how was the sample prepared and spiked with microorganisms. It is not clear in the form performed by the authors. (has been done) both the merge of the section and the describtion of spiked sample as follows: 

Nine hundred and twenty-five grams of cooked ham, purchased in a local market, were divided into 3 portions: i) a portion of 300 g was experimentally contaminated with an overnight culture of L. monocytogenes ATCC 13932 to obtain a final concentration of 104 CFU/g; ii) a portion of 325 g was experimentally contaminated with an overnight culture of L. monocytogenes ATCC 13932 to obtain a final concentration of 106 CFU/g; iii) a portion of 300 g was experimentally contaminated with an overnight culture of Escherichia coli ATCC 25922 and Staphylococcus aureus ATCC 25923 to obtain a final concentration of 104 CFU/g (negative controls). Four different methods were compared to detect L. monocytogenes in all the samples: cultural microbiological method [7], Real- Time PCR [8], Colorimetric LAMP assay, and Real- Time LAMP PCR.

Twenty-five grams of each contaminated portion were placed in sterile bags (Nasco Whirl-Pak, VWR International), gettig 37 samples; in each bag were added 225 mL of sterile non-selective primary enrichment broth (Half Fraser broth, Biolife Italiana S.r.l.).

All the 37 bagged samples were crushed and homogenized in a stomacher blender 400 circular (P.B.I.) and incubated at 30 °C for 24 hours, as required by cultural   microbiological method [7]. After the pre-enrichment step, 2 mL of each sample were subjected to DNA extraction as previously described and detection of L. monocytogenes were carried out as required by colorimetric LAMP, Real-Time PCR and Real-Time LAMP PCR assays. In parallel, all samples were analyzed with cultural microbiological ISO 11290-1 method.

  1. The number of tested samples is very low for comparison of the methods, especially that no differences between efficacy was obtained. Could the authors provide the results from validation of the methods? (The main objective of our work was to verify the performance of the LAMP in RTE food, particularly in cooked ham samples. The comparison with the other methods has been used for to confirm the results obtained by colorimetric LAMP.
  2. According to the authors novelty of the study was based on using LAMP for identification of L. monocytogenesDNA of in raw and ready-to-eat meat. Could the authors compared the efficacy of their “new” colorimetric LAMP with results of Wu et al (2014)? It will be needed in the discussuion part. (has been done by  inserting this sentence in the discussion: "Therefore, with the same efficacy compared to other LAMP assay (13), the    colorimetric LAMP used in this study is more user friendly because no sample handling is  required after DNA amplification". 

Reviewer 2 Report (New Reviewer)

The manuscript is well-written and highly relevant. However, it needs a little work to make it publishable. Please see below the comments. 

Title: As only Cook Ham was used as a ready-to-eat food in this study,  I would suggest replacing the general food category (RTE) with the specific food used (ham). 

Introduction: Please briefly explain the mechanism of the LAMP.

Please remove underlines from the texts. 

L- 87: please remove the space. 

L106-118: Please use the change format of the numbers: 12,5 into 12.5, 2,5 into 2.5 ul, and change it for all. This may confuse the readers. 

Discussion: please write the limitation of the study or the test. 

Author Response

Title: As only Cook Ham was used as a ready-to-eat food in this study,  I would suggest replacing the general food category (RTE) with the specific food used (ham). (has been done)

Introduction: Please briefly explain the mechanism of the LAMP. (has been done)

Please remove underlines from the texts. (has been done)

L- 87: please remove the space. (has been done)

L106-118: Please use the change format of the numbers: 12,5 into 12.5, 2,5 into 2.5 ul, and change it for all. This may confuse the readers. (has been done)

Discussion: please write the limitation of the study or the test. At the end of the discussion we inserted a short sentence about it as follows : "it would be appropriate in the future perform further studies to apply the colorimetric LAMP to detection of L. monocytogenes in naturally contaminated samples as they have a low concentration of the pathogen". 

Reviewer 3 Report (New Reviewer)

This paper is very well written, easy to understand and makes a relevant contribution to the control of Listeria monocytogenes.

The only point that deserves negative mention is the lack of an experiment with naturally contaminated samples, which would close the article with a golden key. Then, I strongly recommend that an experiment with naturally contaminated samples be performed, since even the low concentration used in the experiment may be too high when compared to naturally contaminated samples.

Minor corrections are shown below:

- line 84: replace “ml” by “mL”

- lines 86, 140, 141, 142, 144, 145, 189: replace “μl” by “μL”

- correct the decimal separator "," by "."

- Table 6 isn’t necessary since sensitivity and specificity were total

Author Response

The only point that deserves negative mention is the lack of an experiment with naturally contaminated samples, which would close the article with a golden key. Then, I strongly recommend that an experiment with naturally contaminated samples be performed, since even the low concentration used in the experiment may be too high when compared to naturally contaminated samples.

Given the low probability of finding naturally contaminated samples, we used experimentally contaminate sample; this is also because we wanted to evaluate the test performance in the most favorable conditions.  However, data from our previous studies using experimentally contaminated cooked ham samples have shown that colorimetric LAMP is able to detect L. monocytogenes at concentration level of about 15 – 200 CFU/g.

In addition, all samples used in the study have been analysed for presence of L. monocytogenes,  The result was L. monocytogenes not detectable in 25 g.

Minor corrections are shown below:

- line 84: replace “ml” by “mL” (has been done)

- lines 86, 140, 141, 142, 144, 145, 189: replace “μl” by “μL” (has been done)

- correct the decimal separator "," by "."(has been done)

- Table 6 isn’t necessary since sensitivity and specificity were total (We removed Table 6 fro the text)

Round 2

Reviewer 3 Report (New Reviewer)

The main question has been answered and I agree with the publication of the article as it is.

This manuscript is a resubmission of an earlier submission. The following is a list of the peer review reports and author responses from that submission.

Round 1

Reviewer 1 Report

This is a review of the article 'Development of a Loop-mediated isothermal Amplification (LAMP) assay for the detection of Listeria monocytogenes in Ready-To-Eat meat food'

Title: This needs a little modification. Suggested title is

Application of Loop-mediated isothermal amplification (LAMP) colourimetric assay for the detection of Listeria monocytogenes in Ready-To-Eat meat food'

The reason is that literature is awash with colourimetric LAMP detection 

Line 19: Replace 'shown' with 'show.'

Lines 19-23: This is common knowledge

Figure 3 legends need correction.

Remove panels A and B in Figs 5, 6, and 7. It is not clear even at a magnification of 200x

In Figure 5,6,7, panel c is a table and not a figure. It may be better to just present the tables and present panels A and B as a supplementary figures.

The developmental claims in this article are erroneous because the authors did not develop PCR primers used or the concept of colourimetric LAMP detection. Even the colourimetric master mix used was not assembled by the authors. For developmental studies, the new development should be discussed in detail to show the benefits of the new methods over existing ones. This is lacking in the discussion section. The paper is a straightforward application of existing protocols.

The manuscript should be modified and resubmitted. Proofreading is also required and a  revision of figures and their legends are required.

Reviewer 2 Report

General review:

This communication paper is judged to have industrial value in that it applied a colorimetric LAMP assay that can screen Listeria monocytogenes of RTE-type meat products more rapidly than the existing analysis method.

As intended by the authors, the LAMP assay did not show reaction for any other strains such as Escherichia, Salmonella, and Staphylococcus except for Listeria spp. And also among Listeria spp., other strains except L. monocytogenes did not show the LAMP reaction.

Accordingly, the LAMP assay is confirmed to be an effective method for screening L. monocytogenes for RTE-type meat products.

My comments for improved completeness of this manuscript are as follows:

Title: It is appropriate to classify this paper as communication, so please change “Article” to “Communication” at the top of the title of the manuscript.

Introduction: The definition of “RTE meat food” is ambiguous. What is described in line 35-39 of [Introduction] part is a comprehensive description of RTE-type food products.

Please be more specifically described “RTE meat food”

Materials and Methods: Please specifically described RTE-type meat products used by the authors in the experiment.

e.g., if which was processed meat products, authors must provide the exact food type, such as patties, sausage, or ham.

Results:

1. For the unity of the figures presented by the authors, it is needed to indicate Lane1-9 in Figure 4.

2. It is strongly recommended to improve the resolution of Figures 5-7. It's hard to get the details.

Discussion: I agree that this study provided the LAMP assay can use rapid screening for L. monocytogenes in RTE-type meat products.

However, as the authors described in Line 286-287, the LAMP assay has already been known as a method for screening L. monocytogenes in food.

Regrettably, the authors do not emphasize the novelty of using “RTE-type meat products” as the analytical sample.

Therefore, it is necessary to describe in more detail why the authors detected L monocytogenes in RTE-type meat products using the LAMP assay (this is also the case in the introduction part).

Reviewer 3 Report

The referee understands that the authors are proposing a novel method, but the paper has several drawbacks in the experimental design, lack of statistical analysis, lack of data and method validation and insufficient scientific soundness. The structure of the paper is very weak, and the discussion is very poor.